# A Game-Theoretic Framework to Preserve Location Information Privacy in Location-Based Service Applications

**DOI:** 10.3390/s19071581

**Published:** 2019-04-01

**Authors:** Mulugeta Kassaw Tefera, Xiaolong Yang

**Affiliations:** School of Computer and Communication Engineering, University of Science and Technology Beijing, Beijing 100083, China; mulugetaksw@gmail.com

**Keywords:** mobile networks, location-based service, security and privacy, game-theory, privacy preservation, prisoner’s dilemma, Nash Equilibrium

## Abstract

Recently, the growing ubiquity of location-based service (LBS) technology has increased the likelihood of users’ privacy breaches due to the exposure of their real-life information to untrusted third parties. Extensive use of such LBS applications allows untrusted third-party adversarial entities to collect large quantities of information regarding users’ locations over time, along with their identities. Due to the high risk of private information leakage using resource-constrained smart mobile devices, most LBS users may not be adequately encouraged to access all LBS applications. In this paper, we study the use of game theory to protect users against private information leakage in LBSs due to malicious or selfish behavior of third-party observers. In this study, we model a scenario of privacy protection gameplay between a privacy protector and an outside visitor and then derive the situation of the prisoner’s dilemma game to analyze the traditional privacy protection problems. Based on the analysis, we determine the corresponding benefits to both players using a point of view that allows the visitor to access a certain amount of information and denies further access to the user’s private information when exposure of privacy is forthcoming. Our proposed model uses the collection of private information about historical access data and current LBS access scenario to effectively determine the probability that the visitor’s access is an honest one. Moreover, we present the procedures involved in the privacy protection model and framework design, using game theory for decision-making. Finally, by employing a comparison analysis, we perform some experiments to assess the effectiveness and superiority of the proposed game-theoretic model over the traditional solutions.

## 1. Introduction

Location-based services (LBSs) are becoming increasingly popular, empowering ordinary mobile users with the capacity to access convenience services through wireless networks while, at the same time, sharing location information with possibly untrusted third-party adversarial entities using their Smartphone devices. The key idea behind LBS is to continuously access the users’ private information attached to their locations and service attributes when they interact with such LBS applications. Despite the convenience of LBS applications, many privacy challenges still exist and affect LBS users in various ways. Therefore, existences of vulnerable mobile social network, the lack of users’ awareness of their own privacy protection, and untrusted LBS providers have become the primary factors causing the leakage of users’ personal interests, habits, relationships, and activities [1,2,3,4]. Even mobile users who may question LBS applications’ results may disclose their real locations and potential interests associated with LBSs [5]. According to a survey [6], with the development of mobile social networks and growing use of LBSs, 85% of Internet users are generally worried that their personal data will be acquired and spread by others, which poses a serious threat to personal privacy. There are two main reasons for the leakage of user privacy information in the network: technical defects and interest temptations. At present, researchers have proposed many solutions to protect users’ location privacy, which mainly focused on preventing the leakage of users’ private information through technical means or unwillingly exposing their location to adversarial third-party applications. Recent works [7,8,9] on location privacy have found that a visiting attacker can always access the target’s location history and then launch attacks any time to disclose the correlation between the real identities of the users and their pseudonyms, even when a user’s location information is obfuscated in the presence of location privacy-preserving mechanisms (LPPMs). Therefore, once the location information and user’s identity attached to the LBS queries have been leaked through other techniques or side-channels, users have no means to control their private information, and all aspects of their personal privacy will be revealed. 

To address this problem, a large number of privacy-enhancing mechanisms have been proposed to make use of LBSs, providing them to users, while limiting the amount of sensitive information leaked through observation [10,11,12]. These privacy-preserving mechanisms include location anonymization and user access control mechanisms that attempt to address location privacy issues in location and mobile social networks [13]. Location anonymization is one of the most widely used protection mechanism. It aims to use the anonymous information without using the user’s real location information to obtain the private information of the users [14,15]. The most widely used mechanisms include location obfuscation or spatial-temporal cloaking techniques. However, the protection issue in the anonymization approach is not the possibility of the private information of the owner him/herself being revealed, but the real identity of the private information owner. Therefore, if the real identity information of the user has leaked through the use of LBSs, then all aspects of the user’s privacy will be exposed. On the other hand, the privacy protection mechanism, based on an access control scenario, implements the protection technique from an internal perspective. This mechanism refers to the type of techniques used to appropriately extend the traditional information security-oriented access control mechanisms to achieve the requirements of mitigating various of users’ private information leaking issues [16,17,18]. However, most of these protection mechanisms rely on different technical means in the prevention of the disclosure of a user’s private information and thus has some advantages and disadvantages.

In this paper, we consider the growing use of LBSs to analyze the privacy protection effects from the behavioral interest and game-theoretic perspectives and demonstrate a privacy protection model to be used when third-party visitors access an LBS user’s trajectory. We let the LBSs that enable their users to continuously access them and easily obtain their private information, leaked from their frequently visited routes. Considering the reality that mobile users may not be adequately inspired to access LBSs due to the high risk of privacy using resource-constrained mobile devices, we apply a game-theoretic privacy protection model to analyze the impact of user query behaviors, and identify the Nash equilibrium (NE) solutions. The systems outside the observer could access the user’s trajectory in order to obtain his/her private information through the use of LBSs. When the visitor makes a request to access the user’s private information, the user will decide whether to allow the access query, based on the privacy protection strategy. In this process, it is assumed that both the user and the outsider must pay a certain privacy cost for the possible actions, and the benefits obtained are the factors to be considered and measured in the process of making decisions. Here, we observe the following two important principles [19]: First, different users have different degrees of tolerance for the disclosure of their private sensitive data, reflecting their will to conceal their private information. Second, we suppose that the number of visits by the visitor to the same user has a superposition effect, i.e., as the number of visits to private information increases, the probability that the visitor acquires the user’s private information will also increase and thus the awareness of the user to his/her private information disclosure and protection actions should also increase. 

To analyze the privacy protection problem more effectively, this work is particularly remarkable, because it contains a classic game theory-based privacy protection model that makes it possible for the mobile user and the outside visitor (who aims to obtain the user’s personal data) to compute the game strategies and their corresponding benefits. Based on the benefit, there may be NE in the gameplay between the two parties, and the probability of the visitor’s good-faith access through the equilibrium solution. For this purpose, the user can set a threshold in the privacy protection strategy, according to his or her tolerance of private information leakage. We assume that the visitor is also knowledgeable of this setting of the threshold quality constraint. The game strategies, adopted by both the user and the outsider, can be implemented. From a gaming perspective, the protection model realizes the benefits from the point of view of good-faith or malicious access and allows access to a certain extent (i.e., private information leakage) or denies access (i.e., no leakage). Based on this, the game strategy that can be adopted by the LBS users is “allow access” or “deny access” to their private information, while the visitor adopts a “good-faith access” or “malicious access” strategy in the process of visiting the user’s trajectory. Therefore, the visitor’s access request to the user is allowed only when the probability of the visitor’s good-faith access is higher than the threshold quality constraint. After each visit, the user records the access request involved in the visit as a factor in the revenue calculation of the same visitor’s request in the subsequent game. As the number of visits by the visitor’s access to private information increases, the amount of information the visitor can learn in LBSs may increase according to the superposition effect and the probability of revealing the privacy of the user become larger. Therefore, the same visitor’s request, corresponding to the income received from different visits, is different, and the probability of good-faith access should also decrease until it is lower than the threshold imposed by the user. At this point, the visitor will no longer be allowed to access the private information of the user, according to the privacy protection strategy. This model assumes that both of the game players are rational, and the decision of the visitor and the user is in order, but the strategy adopted by the first decision-maker (i.e., the user) cannot be observed by the visitor, so it can be regarded as both decision-making, and then the game belongs to a static game model. In summary, the contributions of this work are listed as follows:We formally analyze the privacy protection scenarios using game theory and formulate the situation of the prisoner’s dilemma game between a privacy protector (i.e., LBS user) and an outside observer (i.e., user’s private information requester or simply a visitor) in a static game context.We analyze the existence of Nash equilibrium solutions for privacy protection methods and propose a game strategic selection to help users and outsiders obtain the corresponding benefits.We describe the procedures involved in the game model, the gameplay scenario, as well as the framework design used in the game theoretic model.We conduct some experiments based on game theory and the existing technical-oriented solutions, and we then validate the effectiveness of the former through comparison analysis.

The rest of this paper is structured as follows: First, we present an overview of a common LBS system deployment scenario and assumptions in Section 2. In addition, we discuss the adversarial consideration and game-play analysis in the traditional protection model. In Section 3, we discuss the privacy game analysis using the game theory model, so as to obtain the prisoner’s dilemma, faced by both the LBS user and visitor in the traditional protection problems. Moreover, we discuss the detailed game process implementation and the framework design using game theory. Section 4 presents the evaluation analysis and comparative study through experiments to assess the effectiveness and superiority of the proposed game model. Finally, we present related work in Section 5 and summarize this work in Section 6. 

## 2. System Model and Assumptions for LBSs

In this section, we first explain a common LBS architecture based on the probabilistic system framework proposed in [7,20] and its diverse entities, as well as our assumptions. A typical LBS application operates in a centralized fashion, i.e., the location information collected by the mobile devices of users are reported (using wireless communication infrastructures) at a central location server for processing, as illustrated in Figure 1. As shown, the basic architecture of this LBS application consists of the following four major components: (a) LBS users with smartphone devices; (b) positioning services, based on localization infrastructure; (c) communication networks (channels); and (d) third-party service providers. 

During the normal operation of the LBS application, each user needs to report their location-related information to the third-party LBS provider in order to receive the service. The Smartphone devices, used by LBS users, are capable of embedding a variety of positioning systems (e.g., GPS) to identify current geographic locations and other information. Moreover, they can produce a wireless connection through communication channels (e.g., a cellular network), transmit an LBS query and then return the query result from the service providers.

As for the LBS server, the location data are analyzed, associated with nearby points of interest in their databases and made available in various forms, such as a graphical representative set of positions on a map or maps, showing the location trace results at the individual and/or community level. Simultaneously, users’ LBS query results may be displayed locally on their mobile devices or accessed by third-party adversarial entities through mobile and wireless networks, depending on the location information and application needs. Since the location server (such as the LBS provider) is not trusted, users’ sensitive information, attached to their location trace, may be leaked or misused. In order to protect their privacy, users’ usually do not directly interact with an LBS server but prioritize the query by obfuscating the privacy protection mechanism. The privacy of LBS users in a mobile network can be preserved using different protection mechanisms. To simplify privacy protection, the generic LBS adversary model assumes that the third-party service provider is malicious or untrustworthy, while other parts, such as system location privacy-preserving mechanisms (LPPMs), are secure and trusted. For privacy protection purposes, each LBS user’s identity part is normally represented by a pseudonymization in LBS providers, in order to increase the adversary’s uncertainty of the user’s actual whereabouts. These pseudonyms might change at any time, when the users’ access the LBS, or they will keep it for a longer time in order to have a long-term pseudonym. The possibility of changing each pseudonym in accordance with a set of regions depends on the LBS. It is either generated in a distributed architecture or obtained from an offline central authority [10].

### 2.1. Privacy and Adversarial Consideration in LBS

The goal of the adversary is to construct the location access profile of the user’s target and then infer their private information (secret) based on their demographic positions. The adversary can also be the LBS provider itself or untrusted third-party entities in the LBS communication network, capturing the users’ location-related information. As stated before, an adversary may have some side information about the users’ LBS access pattern that aims to disclose their real-time location data and then infer their secret. Therefore, while the users’ access to the various LBSs contributes to real-world mobile network sensing applications, their location privacy may leak in one of three major parts (points), i.e., user’s mobile devices, communication networks and location server by itself [21,22]. According to the LBS architecture, the third-party service provider is generally considered malicious or untrustworthy. After obtaining the user’s movement, an untrusted LBS provider may directly invade the user’s privacy and perform localization attacks or secretly sell his/her private information to other parties [23,24]. 

For example, consider that mobile users use their Smartphone to request a “query to visit the nearest tumor hospital” in order to obtain some services. In this case, the LBS servers receive the LBS query from the user and reply with the query result. Let *U* be the LBS user’s personal sensitive information that the adversary aims to track continuously over space and time and then make illegal benefit from by leveraging this information. We also assume that an adversary *A* can observe all the users’ trajectories in their visits to the nearest tumor hospital. Furthermore, the adversary may also acquire some side information about *U* and may have knowledge of the protection mechanism used by the user. Because users are publicly observable by the third-party LBS provider even if they travel in public places over short periods of time. At one time location, information is identifiable, and this may reveal a detailed profile of the person’s preferences, lifestyle, and habits. We note that the idea behind the traditional protection model design is applicable to LBSs, where users allow his/her trajectory observer to access their private information to a certain extent and where they deny further access when the disclosure of their privacy is about to happen.

### 2.2. Game Analysis in the Traditional Privacy Protection Model

In the traditional game theory model, both of the game players are still the mobile user (i.e., the owner of the location data) and the third-party visitor (who may also be the adversary), who strives to access the user’s private information. The privacy protection model realizes the benefits of both players from the point of view of the “good-faith access’’ or “malicious access” strategy (for the user’s private information visitor) and the “allow access’’ or “deny access to their private information” strategy (for LBS users). In the following, we define the corresponding gain and losses of both privacy game players, when they adopt different strategies:

**D1-**the definition of the user’s income ($Q_{in}{}_{\mathrm{wel}}^{\mathrm{acp}}$): The user’s benefit, obtained by the visitor when an observer adopts a “good-faith access” to the user’s trajectory, while the user adopts the “allow access” strategy. This benefit can also be a loss of the user, when the third-party observer adopts the “good-faith access” strategy, while the user adopts the “deny access to his private information”. On the other hand, it can be seen as having the purpose of providing certain services or expanding the scope of influence by the user by allowing the outsiders to access their private information.

**D2-**visitor’s income ($B_{in}{}_{\mathrm{wel}}^{\mathrm{acp}}$): The visitor’s benefit when a visitor adopts a “good-faith access” strategy, while the user adopts the “allow access” strategy. This benefit can be seen as a visitor acquiring certain services (knowledge) by providing access to the user’s secret in good faith or deepening the knowledge of the LPPM used by the user, so that further communication between both sides can continue.

**D3-**user’s loss ($Q_{loss}{}_{\mathrm{mal}}^{\mathrm{acp}}$): The privacy losses suffered by visitors when a visitor adopts a “malicious access to the user’s trajectory”, while the user adopts the “allow access to his private information” strategy. The loss can be seen as a privacy breach of the user’s location that the adversary has tolerated more than the threshold imposed by the user, causing economic, prestige, career or other harm to the user.

**D4-**visitor’s income ($B_{in}{}_{\mathrm{mal}}^{\mathrm{acp}}$): The visitor’s benefit when an observer adopts a “malicious access to the user’s trajectory”, while the user adopts the “allow access to his private information” strategy. This benefit is different from the gains that the visitors receive through good-faith access and can be seen as the visitor acquiring the extra benefits that he or she expects but that exceed the tolerance of the user’s privacy breach. In general, malicious access takes on a higher privacy risk, and at the same time, the revenue is greater than the benefits that the third-party observers receive through good-faith access.

**D5-**income obtained by the visitor ($Q_{in}{}_{\mathrm{mal}}^{\mathrm{den}}$): The user’s benefit when a visitor adopts a “malicious access to the user’s trajectory”, while the user adopts a “deny access to his private information” strategy. This benefit can be seen as the user successfully protecting his or her private information against a malicious attacker by providing only a probabilistic belief of the user’s private information to the LBS provider. In this case, the LBS user does not want visitors to gain through malicious access. 

Obviously, when both sides of the game players adopt the “good-faith access’’ and the “allow access’’ to the user’s trajectory strategies, they can establish a long-lasting information sharing relationship, which helps the two sides understand each other and cooperate. For instance, when both sides adopt the “malicious access” and the “allow access” strategies, the unilateral trust on the user side provides a convenient condition for the third-party observer to maliciously access the private information that exceeds the tolerance of the threshold imposed by the user. As a result, it brings privacy losses to the service user and additional benefits to the outsider (visitor). On the other hand, when the third-party visitor adopts the “good-faith access to the user’s trajectory’’, while the user takes the “deny access to his private information” strategy, the rejection of the LBS user makes the visitor unable to obtain any revenue, and the user also loses the service due to his own rejection strategy, or the opportunity to expand the influence is also a loss to the LBS user. Furthermore, when the third-party visitor adopts the “malicious access” strategy and the user takes the “deny access to his private information” strategy, the user successfully protects his or her private information, and the visitor does not obtain the user’s secret. As a result, the user obtains a certain benefit, and the visitor did not receive any benefit. Unfortunately, in an open network, the mobile user is unable to take any punitive measures against the third-party visitor’s malicious access behavior.

Based on the above analysis, the reward matrix between the LBS user and third-party visitor in the traditional game theory model can be depicted in Table 1. It indicates that the traditional model only considers the impact of the third-party visitor’s access to the user’s trajectory this time. As a consequence, ignoring the characteristics of the association and superposition of private information obtained by multiple visits, even if it is more the results of a second-time visit may also result in a leak of private information by the visitor that exceeds the tolerance imposed by the user. At the same time, the lack of punishment also makes the users lose the means to protect their private information, so that visitors can make malicious visits without any concern and without worrying about any adverse consequences for themselves.

### 2.3. Reward Matrix for the Traditional Game Theory Model

The goal of the adversarial third-party observer is to identify the real location and pseudonym associations in the LBS query data based on other channel matching or side information and to access the complete sensitive information related to LBSs. A descriptive example is shown in Table 1 to analyze the privacy protection problem between the user and the third-party visitor accessing the user’s private information.

The consequence of the game matrix in Table 1 can be analyzed using the line drawing method. From the perspective of the third-party observer, when the user selects the “allow access to his/her trajectory”, the third-party observer “selfishly or maliciously accesses the user’s secret” and will bring greater benefit to him/herself, i.e., $B_{{\mathrm{in}}_{\mathrm{mal}}^{\mathrm{acp}}}>B_{\mathrm{in}}{}_{\mathrm{wel}}^{\mathrm{acp}}$. At the same time, when the user selects the “deny or refuse access to private information” strategy, the “malicious access” and the “good-faith access to the user’s secret” strategies have the same benefits for the visitor (i.e., both are 0). From the perspective of the LBS user, when the visitor chooses the “good-faith access to the user’s secret”, the “allow access” strategy can bring greater benefit to the user, i.e., $Q_{in_{wel}^{acp}}>-Q_{in_{wel}^{acp}}$. On the other hand, when the visitor chooses a “malicious access” strategy, the “deny access” mechanism can bring greater benefit to the user, i.e., $Q_{loss}{}_{mal}^{den}>-Q_{loss}{}_{mal}^{acp}$. Therefore, the analysis shows that there is a pure NE strategy set in the game matrix ($Q_{\mathrm{loss}}{}_{\mathrm{mal}}^{\mathrm{den}},0$). The pure NE strategy set of the user and the visitor refer to all available sets that they are able to take. In the NE game strategy, this available set contains: “reject access” or “malicious access” to the user’s private information. Let the corresponding game strategy for a player *P_i_* be described as $S_{i}\;\epsilon \;\left\{reject,\;malicious\right\}$.

Obviously, the NE contradicts the basic principle of information exchange and sharing, advocated in the mobile network. Even the location information related to the user’s privacy is not allowed any disclosure but cannot exceed the tolerance of the threshold level imposed by the user’s personal privacy strategy. Therefore, the majority of mobile network users are not always able to choose this balancing strategy. The optimal choice is contrary to the original intention of both players in the game. This is the prisoner’s dilemma between the third-party observer and the LBS user in the traditional game theory model. In the next section, we will use the game theory model to solve the problem of the prisoner’s dilemma game between the third-party observer and the LBS user.

## 3. Privacy Game Analysis using the Game Theory Model

In this section, we adopt the privacy protection scenario based on the game theoretic model to solve the prisoner’s dilemma in the traditional privacy protection model [7,25]. We formulate this game as a non-constant repeated game model to specify the strategic interactions between an outside visitor and privacy protector and provide this game as the relevant theory of Nash equilibrium (NE) solutions. The properties and existence of the NE, with respect to the repeated game contexts, are analyzed in detail. By making use of location information concerning historical access and considering the current access scenario, this game model records the successively visited locations through a feedback mechanism. When the outside observer visits again, the proposed model could realize the benefits based on the past access records (observation history traces) and thus derive the probability that the access is an honest one. The benefit corresponding to the optimal strategy is based on these rewards, and the NE solution is obtained. A choice of strategy by each player is an NE point that is used to obtain the probability of the good-faith access for the visitors. At the same time, the users will set a threshold through the protection strategy based on the levels of tolerance of private information leakage, which is compared with the probability of the visitor making a good-faith visit. The LBS response to location privacy games can be obtained only when the probability of a good-faith visit is higher than the quality threshold set value imposed by the user. As a consequence, the users will adopt the “allow access” strategy, otherwise, they will take the “deny access to their private information” strategy. The privacy protection threshold value is a key parameter that determines whether or not the visitor’s access request is allowed. It can be set by the user, according to his or her privacy requirements or sensitivity. The range value is generally between zero and one inclusive. This threshold value reflects the degree of the user’s tolerance of the privacy breach. Table 2 shows the relationship between the tolerance level and the corresponding threshold values.

When this value is not set, the threshold can be set by default to the average value of 0.5. The LBS response can dynamically set this threshold according to the dynamic state of the network environment and the tolerance of private information leakage. In the process of the visitor’s access to the user’s trajectory, when the likelihood or probability of the good-faith visit is higher than the quality threshold value, the visitor’s request will be allowed. Otherwise, the probability of good-faith access is lower than the threshold, and consequently, the request will be rejected by the user’s privacy protection mechanism.

### 3.1. Game Theory-Based General Privacy Protection Model

Figure 2 illustrates the general process of accessing the user’s private information, in which the proposed game model is used to analyze the protection strategy, based on the observation history traces and the current access scenarios of the whole LBS accessing system. Apparently, when the visitor makes an access query, the accessed location information is analyzed as to whether to allow the query, and an appropriate protection strategy will be applied by users. 

Considering the current and historical access scenarios, the following procedures are available to apply the game theory-based model. (1) The visitor gathers information from users’ trajectories and initiates an access request to visit their private information. (2) The system knows the private information that the visitor requests to access and obtains the private information that the visitor has visited from the historical database. (3) The system combines the various pieces of information about users in step 2 to obtain a set of private information that the visitor can access through the visit, i.e., calculate the superposition effect on the access request. (4) Based on the previous visits, the income corresponding to the different strategies adopted by the visitor and the user is calculated. (5) Based on the different strategies and calculated responses, both the visitor and the user adopt different game strategies that are used to determine the probability of the visitor’s good-faith access. (6) The Nash Equilibrium solution is obtained from the privacy game between the two players. From the NE, the expected value of both players and the probability of selecting each game strategy can be obtained. (7) The proposed model determines whether the access to private information” strategy is an honest one (i.e., the probability of the visitor’s “good-faith access” strategy can be obtained). (8) The system compares the probability that the visitor adopts the “good-faith access to the user’s trajectory” with the access threshold set by the user. If the former is not less than the latter, the “allow access” strategy is adopted. The combination of the private information, obtained by the current access and from the historical observation database, does not exceed the tolerance level of the privacy breach, otherwise, the “refuse access” strategy is adopted. In this case, the visitor could obtain the user’s private information only in the history access records. (9) Finally, the results obtained by multiple visits are recorded in the historical database and applied in subsequent access decisions. When the visitor makes an access request again, the observation history data, recorded in this step, will affect the revenue of both players through steps 2 to 4, thereby affecting the LBS user’s decision to adopt the “allow access” strategy or “reject (deny) access to private information strategy”.

The flow chart in Figure 2 shows that the observation history data, recorded by the access feedback mechanism, and the setting of the threshold value may prove undesirable if the user’s secret is observed by the visitor through the observation and the history traces that have already been obtained. The disclosure of the leaked private information means that the visitor adopts the “malicious access to the user’s private information” and the user will not only disallow the visitor to conduct the observation beyond the private information leakage tolerance but also access the privacy letter according to the LBS provider. The history of interest is a feedback mechanism that imposes penalties or punishments on third-party observers. Therefore, while the two players of the game model issue “malicious access” and “deny access” as their respective strategies, the users required revenue by successfully protecting their private information, and the visitor also lost due to a punishment imposed by the system. In the next section, we describe the game process between the visitor and the user in detail.

### 3.2. The Gameplay Process between the Privacy Protector and the Visitor

In this section, we define the details of the game process between the third-party visitor and the user, with their corresponding benefits. Before the discussion, we first prove the existence of the NE in a finite strategic game. In the game-based privacy protection model, the strategy sets of both players (i.e., the LBS user and visitor) of the game are considered as finite strategy sets, so this is a finite strategic game. According to the NE existence theorem, the game with finite sets of the user and visitor has at least one NE in each finite strategic game [26]. It can be concluded, from this, that there is at least one NE in the game of the privacy protection model. In this finite game model, we use “β” to represent the discounted-payoff value of the user that obtains the future earnings. Similarly, we use “δ” to represent the discounted-payoff value of the visitor that obtains the future earnings. Next, we need to redefine the benefits of both sides under different game strategies.

**D6**-user’s income ($Q_{in}^{\prime }{}_{\mathrm{wel}}^{\mathrm{acp}}$): The benefit of the user when a visitor adopts a “good-faith access”, while the user adopts the “allow access to his/her private information”. This benefit is also the loss of the user, when the visitor chooses the “good-faith access” strategy, while the user adopts the “refuse to access his private information” strategy. Assuming that each time the visitor requests access to the user’s trajectory, the benefit of the user is $q_{\mathrm{in}}{}_{\mathrm{wel}}^{\mathrm{acp}}$, when the visitor requests the user’s private information for the *n*^th^ time:(1)$$Q_{in}^{\prime }{}_{\mathrm{wel}}^{\mathrm{acp}}=q_{\mathrm{in}}{}_{\mathrm{wel}}^{\mathrm{acp}}+q_{\mathrm{in}}{}_{\mathrm{wel}}^{\mathrm{acp}}\times \beta +q_{\mathrm{in}}{}_{\mathrm{wel}}^{\mathrm{acp}}\times \beta^{2}+\cdots +q_{\mathrm{in}}{}_{\mathrm{wel}}^{\mathrm{acp}}\times \beta^{n}=q_{\mathrm{in}}{}_{\mathrm{wel}}^{\mathrm{acp}}\times \frac{1-\beta^{n}}{1-\beta }$$

**D7**-visitor’s income ($B_{in}^{\prime }{}_{\mathrm{wel}}^{\mathrm{acp}}$): The visitor’s revenue when a third-party observer adopts a “good-faith access” strategy, while the user adopts the “allow access” strategy in his trajectory. Assuming that each time the visitor requests access to the private information of the user, the response of the visitor is $b_{in}{}_{wel}^{acp}$, when the visitor requests access to the private information for the *n*^th^ time:(2)$$\begin{array}{c} B_{in}^{\prime }{}_{wel}^{acp}=b_{in}{}_{wel}^{acp}+b_{in}{}_{wel}^{acp}\times \delta +b_{in}{}_{wel}^{acp}\times \delta^{2}+\cdots +b_{in}{}_{wel}^{acp}\times \delta^{n}\\ \;=b_{in}{}_{wel}^{acp}\times \frac{1-\delta^{n}}{1-\delta } \end{array}$$

**D8**-user’s loss ($Q_{\mathrm{loss}}^{\prime }{}_{\mathrm{mal}}^{\mathrm{acp}}$): The visitor’s benefit when an observer adopts a “malicious access” strategy, while the user adopts the “allow access” strategy. Assume that each time the visitor requests access to the users, the visitor’s income is $q_{\mathrm{loss}}{}_{\mathrm{mal}}^{\mathrm{acp}}$, and then when the visitor requests access to the private information for the *n*^th^ time.(3)$$Q_{\mathrm{loss}}^{\prime }{}_{\mathrm{mal}}^{\mathrm{acp}}=q_{\mathrm{loss}}{}_{\mathrm{mal}}^{\mathrm{acp}}+q_{\mathrm{loss}}{}_{\mathrm{mal}}^{\mathrm{acp}}\times \beta +q_{\mathrm{loss}}{}_{\mathrm{mal}}^{\mathrm{acp}}\times \beta^{2}+\cdots +q_{\mathrm{loss}}{}_{\mathrm{mal}}^{\mathrm{acp}}\times \beta^{n}=q_{loss}{}_{mal}^{acp}\times \frac{1-\beta^{n}}{1-\beta }$$

**D9**-visitor’s income ($B_{in}^{\prime }{}_{\mathrm{mal}}^{\mathrm{acp}}$): The visitor’s revenue when an observer adopts a “malicious access” strategy, while the user adopts the “allow access” strategy. Assuming that each time the visitor requests access to the user’s trajectory, the benefit of the visitor is $b_{\mathrm{in}}{}_{\mathrm{mal}}^{\mathrm{acp}}$, when the visitor requests access to the user’s secret for the *n*^th^ time.(4)$$B_{in}^{\prime }{}_{\mathrm{mal}}^{\mathrm{acp}}=b_{\mathrm{in}}{}_{\mathrm{mal}}^{\mathrm{acp}}+b_{\mathrm{in}}{}_{\mathrm{mal}}^{\mathrm{acp}}\times \delta +b_{\mathrm{in}}{}_{\mathrm{mal}}^{\mathrm{acp}}\times \delta^{2}+\cdots +b_{\mathrm{in}}{}_{\mathrm{mal}}^{\mathrm{acp}}\times \delta^{n}=b_{in}{}_{mal}^{acp}\times \frac{1-\delta^{n}}{1-\delta }$$

**D10**-visitor’s income $\left(Q^{\prime }{}_{{\mathrm{in}}_{\mathrm{mal}}^{\mathrm{den}}}\right)$: The visitor’s revenue when an observer adopts a “malicious access” strategy, while the user takes a “deny access” strategy. Assuming that each time the visitor requests access to users, the visitor’s revenue is $q_{{\mathrm{in}}_{\mathrm{mal}}^{\mathrm{den}}}$, when the visitor requests access to the user’s secret for the *n*^th^ time.(5)$${Q^{\prime }}_{{\mathrm{in}}_{\mathrm{mal}}^{\mathrm{den}}}=q_{{\mathrm{in}}_{\mathrm{mal}}^{\mathrm{den}}}+q_{{\mathrm{in}}_{\mathrm{mal}}^{\mathrm{den}}}\times \beta +q_{{\mathrm{in}}_{\mathrm{mal}}^{\mathrm{den}}}\times \beta^{2}+\cdots +q_{{\mathrm{in}}_{\mathrm{mal}}^{\mathrm{den}}}\times \beta^{n}=q_{in_{mal}^{den}}\times \frac{1-\beta^{n}}{1-\beta }$$

**D11**-visitor’s loss $\left(B^{\prime }{{}_{\mathrm{loss}}}_{\mathrm{mal}}^{\mathrm{den}}\right)$: The visitor’s loss when an outsider adopts “malicious access” and “deny access” strategies at the same time. Assuming that each time the visitor requests access to the user, the visitor’s response is $b_{\mathrm{loss}}{}_{\mathrm{mal}}^{\mathrm{den}}$, when the visitor requests access to the user for the *n*^th^ time. (6)$$B_{\mathrm{loss}}^{\prime }{}_{\mathrm{mal}}^{\mathrm{den}}=b_{\mathrm{loss}}{}_{\mathrm{mal}}^{\mathrm{den}}+b_{\mathrm{loss}}{}_{\mathrm{mal}}^{\mathrm{den}}\times \delta +b_{\mathrm{loss}}{}_{\mathrm{mal}}^{\mathrm{den}}\times \delta^{2}+\cdots +b_{\mathrm{loss}}{}_{\mathrm{mal}}^{\mathrm{den}}\times \delta^{n}=b_{loss}{}_{mal}^{den}\times \frac{1-\delta^{n}}{1-\delta }$$

Based on the expected gain and loss analysis of the user and visitor, shown above, the game matrix for both players is shown in Table 3. 

The reward matrix in Table 3 can be analyzed using the line drawing method. From the perspective of the third-party observer, when the user selects the “allow access” strategy, the “malicious access” strategy is adopted by the visitor. This brings greater benefit to the visitor, i.e., $B^{\prime }{{}_{in}}_{mal}^{acp}>{B^{\prime }}_{in}{}_{wel}^{acp}$. Similarly, when the user selects the “deny access” strategy, the benefits of adopting the “malicious access” strategy for the visitor is less than the benefit of adopting the “good-faith access” strategy, i.e., $-B^{\prime }{{}_{loss}}_{mal}^{den}<0$. From the perspective of the LBS user, when the visitor chooses the “good-faith access” strategy, the “allow access” strategy can bring extra benefits to the user, i.e., ${Q^{\prime }}_{in_{wel}^{acp}}>-Q^{\prime }{}_{in_{wel}^{acp}}$. Similarly, when a third-party LBS observer chooses the “malicious access” strategy, the “deny access” strategy can bring more benefits to the user, i.e., $Q^{\prime }{}_{in_{mal}^{den}}>-{Q^{\prime }}_{loss}{}_{mal}^{acp}$. From the overall analysis, we find that there is no pure strategy NE point in the reward matrix. Therefore, we need to calculate its mixed strategy NE (randomization) solution. The randomized or mixed strategies allow each player to choose a probability of playing each allowable strategy.

We use $Pr_{B}$ and $Pr_{Q}$ to represent the reward (income) matrix of the visitor and the user in Table 4. Assuming that the probability of the user selecting the “allow access” strategy is x, and the “deny access” strategy is 1 − x, the probability of the mixed strategy for the user is ${\mathrm{Pr}}_{Q}=\left(x,1-x\right)$. Similarly, assuming that the probability of the third-party LBS observer selecting a “good-faith access” strategy is y and a “malicious access” strategy is 1 − y, the probability of a visitor’s hybrid strategy is ${\mathrm{Pr}}_{B}=\left(y,1-y\right)$. Now, we can assume that there are N-players in the n-user game. The game’s income function $E_{Q}$ of the user can be determined by Equation (7), shown below:(7)$$\begin{array}{c} E_{Q}={\mathrm{Pr}}_{q}\times {\mathrm{Pr}}_{Q}\times {\mathrm{Pr}}_{b}^{T}=[x\;1-x][\begin{array}{cc} Q^{\prime }{}_{{\mathrm{in}}_{\mathrm{wel}}^{\mathrm{acp}}} & -Q^{\prime }{}_{{\mathrm{loss}}_{mal}^{\mathrm{acp}}}\\ -Q^{\prime }{}_{{\mathrm{in}}_{\mathrm{wel}}^{\mathrm{acp}}} & Q^{\prime }{}_{{\mathrm{in}}_{\mathrm{mal}}^{\mathrm{den}}} \end{array}][\begin{array}{c} y\\ 1-y \end{array}]\\ \;=(2x-1)\times y\times Q^{\prime }{}_{{\mathrm{in}}_{\mathrm{wel}}^{\mathrm{acp}}}+x\times (-Q^{\prime }{}_{{\mathrm{loss}}_{\mathrm{mal}}^{\mathrm{acp}}})\times (1-y)+(1-x)\times (1-y)\\ \;\times Q^{\prime }{}_{{\mathrm{in}}_{\mathrm{mal}}^{\mathrm{den}}} \end{array}$$

Here, $E_{Q}$ reduces x, and we can get Equation (8) as follows:(8)$$\frac{\partial E_{Q}}{\partial x}=2\times y\times {Q^{\prime }}_{\mathrm{in}}{}_{\mathrm{wel}}^{\mathrm{acp}}-\left({Q^{\prime }}_{\mathrm{loss}}{}_{\mathrm{mal}}^{\mathrm{acp}}+{Q^{\prime }}_{\mathrm{in}}{}_{\mathrm{mal}}^{\mathrm{den}}\right)\times \left(1-y\right)$$

Assuming that Equation (8) is equal to 0, and the value of y can be obtained, Equation (9) can be expressed as follows:(9)$$y=\frac{\frac{{Q^{\prime }}_{\mathrm{loss}}{}_{\mathrm{mal}}^{\mathrm{acp}}+{Q^{\prime }}_{\mathrm{in}}{}_{\mathrm{mal}}^{\mathrm{den}}}{2\times {Q^{\prime }}_{\mathrm{in}}{}_{\mathrm{wel}}^{\mathrm{acp}}+{Q^{\prime }}_{\mathrm{loss}}{}_{\mathrm{mal}}^{\mathrm{acp}}+{Q^{\prime }}_{\mathrm{in}}{}_{\mathrm{mal}}^{\mathrm{den}}}+q_{\mathrm{loss}}{}_{\mathrm{mal}}^{\mathrm{acp}}\times \frac{1-\beta^{n}}{1-\beta }+q_{\mathrm{in}}{}_{\mathrm{mal}}^{\mathrm{den}}\times \frac{1-\beta^{n}}{1-\beta }}{(2\times q_{\mathrm{in}}{}_{\mathrm{wel}}^{\mathrm{acp}}\times \frac{1-\beta^{n}}{1-\beta }+q_{\mathrm{loss}}{}_{\mathrm{mal}}^{\mathrm{acp}}\times \frac{1-\beta^{n}}{1-\beta }+q_{\mathrm{in}}{}_{\mathrm{mal}}^{\mathrm{den}}\times \frac{1-\beta^{n}}{1-\beta }}=\frac{q_{\mathrm{loss}}{}_{\mathrm{mal}}^{\mathrm{acp}}+q_{\mathrm{loss}}{}_{\mathrm{mal}}^{\mathrm{den}}}{q_{\mathrm{in}}{}_{\mathrm{wel}}^{\mathrm{acp}}+q_{\mathrm{loss}}{}_{\mathrm{mal}}^{\mathrm{acp}}+q_{\mathrm{in}}{}_{\mathrm{mal}}^{\mathrm{den}}}$$

Similarly, the visitor’s response $E_{B}$ can be calculated by Equation (10) as follows:(10)$$\begin{array}{c} E_{B}={\mathrm{Pr}}_{q}\times {\mathrm{Pr}}_{B}\times {\mathrm{Pr}}_{b}^{T}=[x\;1-x][\begin{array}{cc} B^{\prime }{}_{{\mathrm{in}}_{\mathrm{wel}}^{\mathrm{acp}}} & B^{\prime }{}_{{\mathrm{in}}_{\mathrm{mal}}^{\mathrm{acp}}}\\ 0 & -B^{\prime }{}_{{\mathrm{loss}}_{\mathrm{mal}}^{\mathrm{den}}} \end{array}][\begin{array}{c} y\\ 1-y \end{array}]\\ \;=x\times B^{\prime }{}_{{\mathrm{in}}_{\mathrm{wel}}^{\mathrm{acp}}}\times y+x\times B^{\prime }{}_{{\mathrm{in}}_{\mathrm{mal}}^{\mathrm{acp}}}\times (1-y)+(1-x)\times (-B^{\prime }{}_{{\mathrm{loss}}_{\mathrm{mal}}^{\mathrm{den}}})(1-y) \end{array}$$

$E_{B}$ derives y, and we can get Equation (11) as follows:(11)$$\frac{\partial E_{B}}{\partial y}=x\times \left({B^{\prime }}_{\mathrm{in}}{}_{\mathrm{wel}}^{\mathrm{acp}}-{B^{\prime }}_{\mathrm{in}}{}_{\mathrm{mal}}^{\mathrm{acp}}-B^{\prime }{{}_{\mathrm{loss}}}_{\mathrm{mal}}^{\mathrm{den}}\right)+{B^{\prime }}_{\mathrm{loss}}{}_{\mathrm{mal}}^{\mathrm{den}}$$(12)$$\begin{array}{c} x=\frac{{B^{\prime }}_{\mathrm{loss}}{}_{\mathrm{mal}}^{\mathrm{den}}}{{B^{\prime }}_{\mathrm{in}}{}_{\mathrm{wel}}^{\mathrm{acp}}+{B^{\prime }}_{\mathrm{loss}}{}_{\mathrm{mal}}^{\mathrm{den}}-{B^{\prime }}_{\mathrm{in}}{}_{\mathrm{wel}}^{\mathrm{acp}}}\\ \;=\frac{b_{\mathrm{loss}}{}_{\mathrm{mal}}^{\mathrm{den}}\times \frac{1-\delta^{n}}{1-\delta }}{\left(b_{\mathrm{in}}{}_{\mathrm{mal}}^{\mathrm{acp}}\times \frac{1-\delta^{n}}{1-\delta }+b_{\mathrm{loss}}{}_{\mathrm{mal}}^{\mathrm{den}}\times \frac{1-\delta^{n}}{1-\delta }-b_{\mathrm{in}}{}_{\mathrm{wel}}^{\mathrm{acp}}\times \frac{1-\delta^{n}}{1-\delta }\right)}\\ \;=\frac{b_{\mathrm{loss}}{}_{\mathrm{mal}}^{\mathrm{den}}}{b_{\mathrm{in}}{}_{\mathrm{mal}}^{\mathrm{acp}}+b_{\mathrm{loss}}{}_{\mathrm{mal}}^{\mathrm{den}}-b_{\mathrm{in}}{}_{\mathrm{wel}}^{\mathrm{acp}}} \end{array}$$

Therefore, the expected mixed strategy result of the NE solution is shown in Equation (13):(13)$$\begin{array}{c} \left[\begin{array}{cc} x & 1-x \end{array}\right]=\left[\begin{array}{cc} \frac{b_{\mathrm{loss}}{}_{\mathrm{mal}}^{\mathrm{den}}}{b_{\mathrm{in}}{}_{\mathrm{mal}}^{\mathrm{acp}}+b_{\mathrm{loss}}{}_{\mathrm{mal}}^{\mathrm{den}}-b_{\mathrm{in}}{}_{\mathrm{wel}}^{\mathrm{acp}}} & 1-\frac{b_{\mathrm{loss}}{}_{\mathrm{mal}}^{\mathrm{den}}}{b_{\mathrm{in}}{}_{\mathrm{mal}}^{\mathrm{acp}}+b_{\mathrm{loss}}{}_{\mathrm{mal}}^{\mathrm{den}}-b_{\mathrm{in}}{}_{\mathrm{wel}}^{\mathrm{acp}}} \end{array}\right],\\ \left[\begin{array}{cc} y & 1-y \end{array}\right]=\left[\begin{array}{cc} \frac{q_{\mathrm{loss}}{}_{\mathrm{mal}}^{\mathrm{acp}}+q_{\mathrm{in}}{}_{\mathrm{mal}}^{\mathrm{den}}}{2\times q_{\mathrm{in}}{}_{\mathrm{wel}}^{\mathrm{acp}}+q_{\mathrm{loss}}{}_{\mathrm{mal}}^{\mathrm{acp}}+q_{\mathrm{in}}{}_{\mathrm{mal}}^{\mathrm{den}}} & 1-\frac{q_{\mathrm{loss}}{}_{\mathrm{mal}}^{\mathrm{acp}}+q_{\mathrm{in}}{}_{\mathrm{mal}}^{\mathrm{den}}}{2\times q_{\mathrm{in}}{}_{\mathrm{wel}}^{\mathrm{acp}}+q_{\mathrm{loss}}{}_{\mathrm{mal}}^{\mathrm{acp}}+q_{\mathrm{in}}{}_{\mathrm{mal}}^{\mathrm{den}}} \end{array}\right] \end{array}$$

From the mixed strategy NE solution, shown above, we can understand that the protection model can obtain the probability that, when the LBS user selects the “allow access” strategy, the visitor will choose the “good-faith access” strategy in relation to the user’s private information. Since the user sets an access threshold in the LPPM, according to the degree of tolerance of the user to the private information leakage, the probability that the visitor chooses the “willingly access the private information” strategy is compared with the access threshold. If the probability is not less than the threshold, the user allows the access request, otherwise, he/she rejects the visitor’s access request.

### 3.3. Design Framework for the Game-Theoretic Protection Model

In this section, we illustrate the general framework design of game theory-based privacy protection model and point out its main components. After analyzing the game process and corresponding benefits to each player, Figure 3 illustrates a concrete design framework for the privacy protection model. In general, the designed framework consists of creating the LBS applications according to the requirements, such as executing several access tasks and actions (e.g., collecting, and controlling) on the mobile device of an LBS user, computing and reporting decisions, and storing the observed data. Inspired by the general trust model for the decision-making framework, proposed in [25], we further divide the architectural design of the game model into three phases: user’s LBS access task execution, private information decision-making system, and observation history access database, as shown in Figure 3. The task execution phase mainly includes collecting the basic access information to third-party observers on the mobile device of an individual user and executing the access control decision of LBS users. As shown, the task execution part mainly includes an information acquisition module and a decision execution module. The private information acquisition module is responsible for collecting the user’s location and private information requested by the visitor and providing the information to the decision-making part. The decision execution module is responsible for receiving the final decision result of the feedback for making and executing decisions. 

The decision-making phase mainly includes receiving the basic access information, collected in the execution phase, calculating the access control decision result through a series of related algorithms and submitting the result to the execution section, and recording the private information observed by the visitor in the observation history access database. The game theory decision-making part includes three sub-modules: information gathering and collection module, game-processing module, and threshold comparison module. The private information gathering and collection module is responsible for receiving observed information, provided by the execution part, and the observation history trace, accessed by the visitor in the historical access database and combining them to calculate the visitor through the access visits. The game-process module is responsible for calculating the mixed strategy NE in the game between the third-party visitor and the user and then calculating the probability of the visitor choosing the “good-faith visit” strategy through the mixed NE strategy. The threshold comparison module is responsible for comparing the probability that the visitor, obtained by the game module, adopts the “good-faith access” strategy with the threshold set by the user in advance, and if the former is not less than the latter, the “allow access to private information” strategy is chosen. Otherwise, the “reject access” strategy is implemented, and the decision is provided to the execution module. Finally, the observation history access database is mainly responsible for storing the user’s private data, observed by the visitor, for use in decision-making and the calculation of the access control decision.

## 4. Experimental Analysis and Discussion

This section presents the experimental analysis to assess the performance of the proposed game theory-based privacy protection model and traditional access control solutions by leveraging the realistic mobility datasets. In Section 4.1, we present some theoretical discussions on existing traditional privacy protection mechanisms in comparison with the game theory-based protection model. Our evaluation results are demonstrated against a random approach, where the owner of the private information randomly decides whether to allow the request of the visitor to access their private information to some extent. The evaluation results and discussions of this analysis are presented in Section 4.2.

### 4.1. The Scenario Settings of the Evaluation System

In this paper, we take a different approach, studying privacy protection from the perspective of behavioral interest and the game-theoretic model, and we establish a privacy protection mechanism in LBS applications. The visited user’s trajectory could be used to identify the users’ exact positions and their private information in the other publicly available datasets. The existing research works demonstrate that privacy protection mechanisms, based on technical-oriented mechanisms (collectively referred to herein as traditional models) [13,16,17,18,27,28,29], merge analogous privacy protection strategies with existing information security-oriented access control models. The existing access control strategy reflects the necessity of privacy protection and information security techniques and then use the traditional user access control strategies, according to the identity, role, attribute or contexts, to authorize access requests, established access control policies, and the making of decisions that allow or refuse access to private information. The particular process of empowering the visitor’s access inquiry still comes after the outside visitor requests to access the private information, and the accessed object (the owner) controls the request based on the privacy protection strategies. Therefore, these traditional privacy protection mechanisms based on the access control model naturally inherit the feature of independently authorizing each access inquiry and usually ignore the important requirements of privacy protection based on the features of preventing visitors from accessing users’ information many times. The visited user’s trajectory is superimposed to ultimately obtain the private information, and the user dynamically develops a privacy protection strategy in real time to ensure that the private information acquired by the visitor does not exceed the level of personal private information leakage tolerated by the visitors. The difference between these protection models and the game-theoretic approach is that the traditional solution usually ignores other important characteristics of privacy protection and only decides whether the visitor’s current access inquiry is a malicious or honest one. From the game-theory perspective, the strategies adopted by the privacy game players and the corresponding benefits are analyzed to establish a privacy game model, and finally, the protection model adopts the visitor’s good-faith access in agreement with the threshold value imposed by the user.

### 4.2. Data Set and Evaluation Results

In this experiment, we use the real mobility dataset from the Geolife project [30,31] to simulate the proposed game theory-based model and traditional access control solutions. This dataset contains a large number of moving trajectories of mobile users (i.e., around 17,621 identified trajectories). They were collected from 182 users over three years. In this analysis, we randomly choose 20 self-interested LBS users and 100 outsiders to visit the users’ trajectories. The users randomly tag multiple pieces of private information as sensitive personal data, i.e., not expected to be leaked. When the third-part visitor successfully accesses private information that exceeds the degree of tolerance of the user, it will cause all aspects of private information about the users to be leaked. In the first experimental analysis, we observe that the visitors use the traditional and game theory-based privacy protection models to examine the user’s private information and the effectiveness of them for privacy protection. The characteristics of both protection solutions and the comparison analysis are shown in Figure 4, Figure 5 and Figure 6, respectively. Hence, in order to obtain equitable evaluation results, we ask the outside visitor to randomly observe the user for 100 successive visits in a row, and we randomly extract 30 successive visits to analyze the privacy protection effects of a certain collection of the user’s private information. Moreover, we repeat this analysis around 50 times and then extract the average of these 50 evaluation results as the final experimental outputs. Table 4 shows the relationship between the user’s private information leakage probability and the number of visits in the experiment. We estimate the validity of privacy protection using Equation (14), where ρ is the probability of privacy breach, and ϵ is the validity of privacy protection.(14)$$\epsilon ={\left(\frac{1}{2}\right)}^{\rho -1}-1$$

The results are shown in Figure 4. As can be seen, as the number of location access increases, the traditional and game theory-based privacy protection models increase the leakage of private information probability. 

This is due to the increasing number of successive visits and the additive effect of the visited user’s trajectory contents, i.e., as the number of visits increases, the probability of the user’s private information leakage will increase. The level of privacy breach is tolerated, but the personal privacy revealed by combining them may also be greater than the tolerance of the user. However, in comparison with the game theory-based protection model, the probability of private information leakage in the game theory model is always lower than in the traditional privacy protection model. As shown in Figure 4, the probability of leaked private information in the traditional protection mechanism is always greater than in the proposed game-based protection model. In addition, in order to observe the relationship between the thresholds set by the user and the number of visits in the game-based privacy protection model, we still assume that there are 100 visitors and 20 users and then set the visitors for 100 consecutive random accesses. We randomly extract 70 consecutive visits to observe the number of visits by which the visitor exceeded the user’s tolerance for privacy breaches of different thresholds. The threshold here refers to a value set by the user according to his tolerance of private information leakage. Different users can set different thresholds according to their tolerance of the leakage of different aspects of personal information. The “allow access to private information” strategy will be taken when the probability of a “good-faith access” strategy is above this threshold. To further ensure the accuracy and objectivity of the experimental analysis, we repeat the evaluation around 50 times and take 50 average results as the final experimental output. Table 5 shows the relationship between the threshold set by the user and the number of access results. The results are shown in Figure 5, where the effectiveness of privacy protection between the traditional and game theory-based model decreases with the increase in the number of accesses. This is also the inevitable result of the superposition effect of repeated access, i.e., as the number of visitor accesses increases, the private data acquired by the visitor will be more and more. As shown in Figure 5, the effectiveness of the traditional protection model to preserve the user’s private data is reduced, while the game-based model protects the private information of the user and always better than the former.

Furthermore, another feature of the game theory-based protection model is the flexible setting of the privacy threshold imposed by the user. The user sets the sensitivity or the privacy information protection requirement to decide whether to allow or deny the visitor to access his/her information. This privacy threshold value is used to compare with the visitor’s probability to adopt the ‘‘good-faith access to the user’s private information” strategy. The “allow access to the user’s trajectory” strategy is adopted only when the probability of the visitor’s good-faith access is not less than the access threshold imposed by the user, otherwise, the “refuse access” strategy is adopted. Figure 6 shows that in the game-based privacy protection model, as the threshold increases, the number of visits to the user’s tolerance of private information leakage is reduced, i.e., the audience is less tolerant of private information leakage. Conversely, the lower the threshold, the more visitors’ good-faith access will not exceed the number of visits to protect the user’s privacy, i.e., the user’s tolerance of private information leakage is higher.

## 5. Related Works

Preserving location privacy in LBSs has attracted significant research attention, supporting a wide range of applications since their emergence in recent years. Smartphone users need to share their personal information with possibly untrusted third-party visitors or service providers in various LBSs to receive some services. In these applications, we differentiate among scenarios based on the frequency at which the mobile users access a service through mobile and wireless networks. On the one hand, there are some LBS applications that require the mobile users to access them continuously (rather than sporadically), whereas, on the other hand, the majority of LBSs have users disclose their information sporadically (rather than continuously). In these cases, there are two successive accesses of a user to the LBS access pattern, with non-negligible gaps in time. Most of the existing LBS applications, such as services that allow for nearby points-of-interest, local events or near-by friends to be identified, are sporadic, as the LBSs do not need their users to expose their location all the time [32]. The authors in [33] pointed out that a sequence of location sharing services are sporadic, i.e., sparsely distributed over time, and the adversary can identify a person over space and time. In addition, several research works have studied that location privacy, even user location data under the protection mechanism, are subject to erroneous or malicious attacks [33,34,35,36,37]. Specifically, with the aid of other side channels, a malicious visitor attack could identify the users’ present, past, and future locations, and then infer their whereabouts in order to act to damage their well-being [7,8]. In this paper, we focus on different approaches to the study of the privacy protection issues against outside visitors who have historical access data and malicious visits.

The location of the user is available in a spatial as well as temporal form. For instance, it is possible for a visitor to obtain access to a user’s current location and history traces, which is called temporal access. In other scenarios, spatial access is considered critical if the user’s position is located geographically. For instance, the leak of the user’s identified location over time, along with their identity, would expose their behaviors in time and their interest in places by linking and inference attacks. Hence, these spatial and temporal resolutions, associated with the position and time of the user, are important parameters for defining location privacy. The most popular approach to achieving k-anonymity location privacy in LBS is either utilizing location obfuscation or spatial-temporal cloaking techniques [38,39,40]. However, the spatial-temporal cloaking approach is not appropriate for the application scenario, when users use the same pseudonym locations to a certain extent. On the other hand, location obfuscation mechanisms act as a noisy information channel between the LBS provider and the user’s actual location, with the intention of confusing an adversary [41]. According to [42,43], when users want to protect their location privacy in LBSs, they include false or fixed location information, rather than the actual locations of LBS users, as service query parameters. The approaches of many works, such as [44,45,46], fall into this category, validating the k-anonymity privacy protection effects. Both location obfuscation and spatial-temporal cloaking techniques may impair the user’s private information and quality of LBS due to reporting of the obfuscated or coarse-grained version of the location information. In this paper, we let the LBSs that enable their users to continuously access their locations to obtain their private information leakage in LBS while analyzing the privacy protection effects from the perspective of game theory and incentives. We note that the principle behind our proposed framework design is applicable to LBSs, where users dynamically develop a privacy protection strategy in order to ensure that the private information, acquired by their potential visitor does not exceed the level of private information leakage tolerated by the visitors.

Game theory is an advanced study of strategic interaction between self-interested individuals (i.e., game players). Due to the high cost of privacy risk and resource consumption issues, self-interested mobile users may not be sufficiently motivated to access all the LBS applications. In the location-privacy scenario, the game theory model is an advanced tool for analyzing the user’s privacy protection problems with various LBSs [7,47,48]. In this paper, we analyze the existing technical-oriented mechanisms to preserve the privacy and protect the interests of a user that utilizes LBSs against potential intruders. In contrast to existing solutions, we propose a framework, based on and inspired by the field of game theory, to better understand the intrinsic nature of privacy protection issues. A similar approach is used by [7] in the design of a distributed dummy user generation, taking into account the non-cooperative behavior of self-interested mobile users to identify the equilibrium solutions. Our solution is different from this previous work on protecting users’ private information issues. In our study, we demonstrate a privacy protection model that illustrates a procedure involved in the model, the scenario of gameplay between a privacy protector and third-party visitor, as well as a framework design of a game theory model. 

## 6. Conclusions

With the wide use of LBS applications, more and more private information regarding users’ locations are collected and shared with untrusted third-party visitors, which has raised serious privacy issues due to malicious visits. In this paper, we first discussed the existing technical-oriented mechanisms to preserve the privacy of an LBS user and analyzed their characteristics and those of different technical approaches in the prevention of user privacy information. Then, we studied the prevention of a user’s private information leakage in LBSs from the perspective of the behavioral interest of mobile users and game theory. From this, we proposed a privacy protection model based on game theory to better understand the intrinsic nature of the user’s location privacy protection strategy. Our proposed framework shall help users to deal with privacy concerns by giving LBS users and their potential opponents a way to negotiate with one another in relation to the access of the private information of the user based on gaming aspects and incentives. From the perspective of game theory, we realized the strategy of privacy protection by disclosing the user’s trajectory to a certain extent, while denying access in cases where the disclosure of privacy was about to happen. By considering the current access scenario and making use of collected private information concerning historical access traces, the game theoretic model derives the probability that the visitor’s access is honest. In this context, traditional game theory aspects, such as the existence of a Nash Equilibrium solution in a finite strategic game, were analyzed. Based on this, we further illustrated the scenario of the gameplay between the visitor and the privacy protector as well as a framework design of the game-theoretic model. Experimental analysis shows that the proposed game-theoretic protection model can protect user privacy better than the conventional privacy protection models.

## Figures and Tables

**Figure 1 sensors-19-01581-f001:**
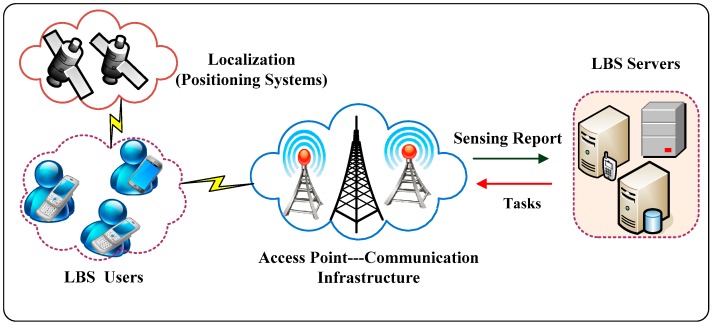
The common location-based service (LBS) system deployment scenario: LBS servers, localization, communication networks, and mobile users.

**Figure 2 sensors-19-01581-f002:**
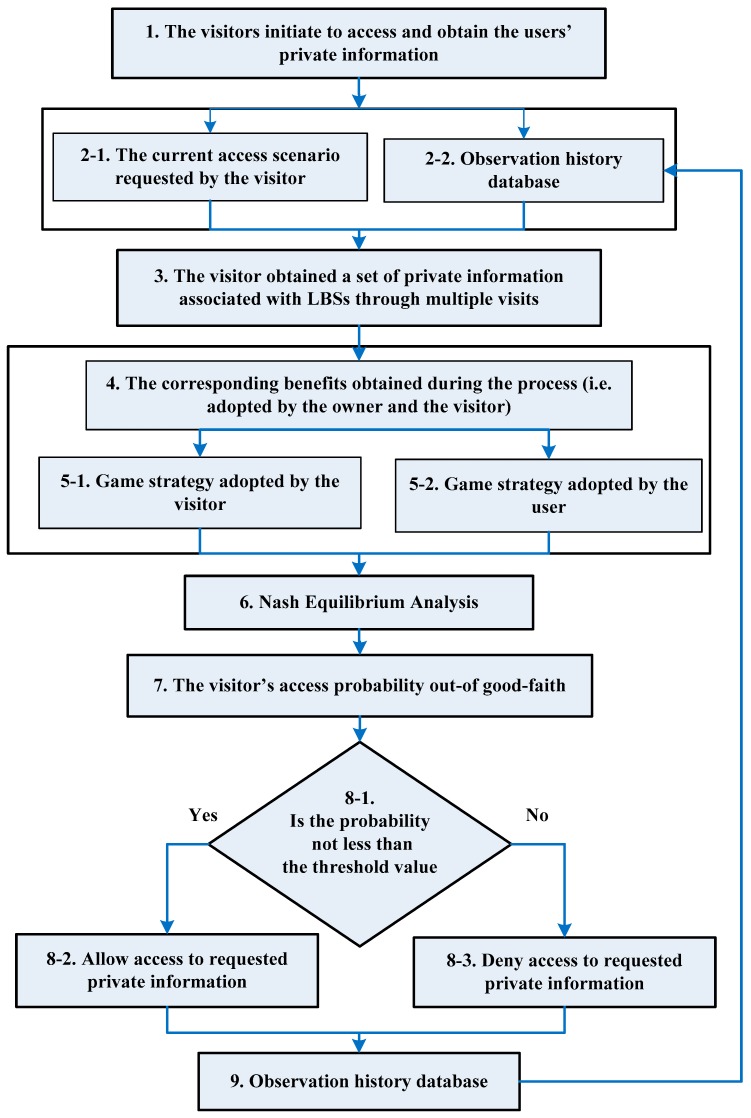
Proposed game-theory privacy protection model.

**Figure 3 sensors-19-01581-f003:**
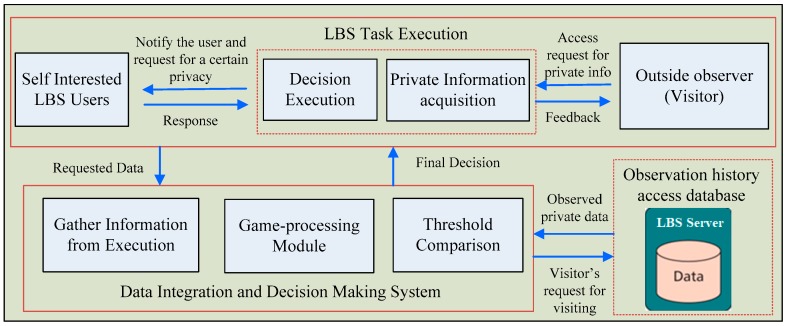
A Framework Design of the game-theory based privacy protection model.

**Figure 4 sensors-19-01581-f004:**
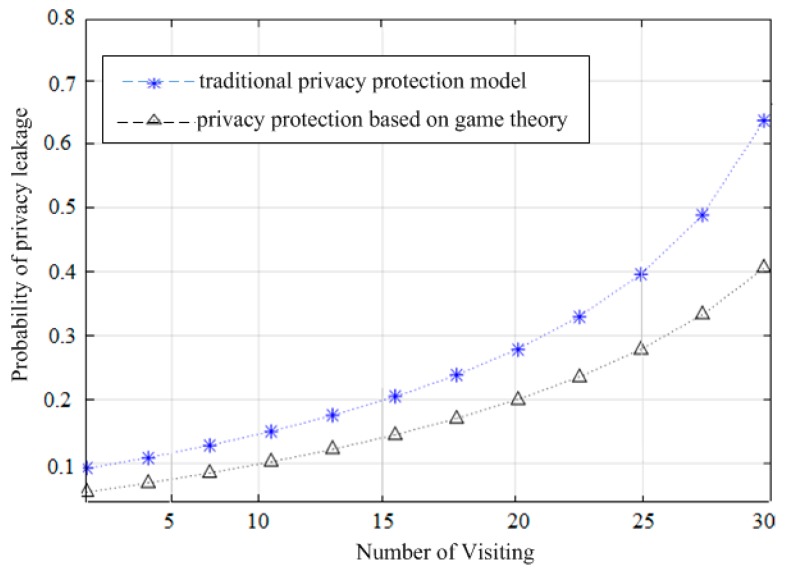
The probability of private information leakage between the traditional and game theory-based privacy protection models.

**Figure 5 sensors-19-01581-f005:**
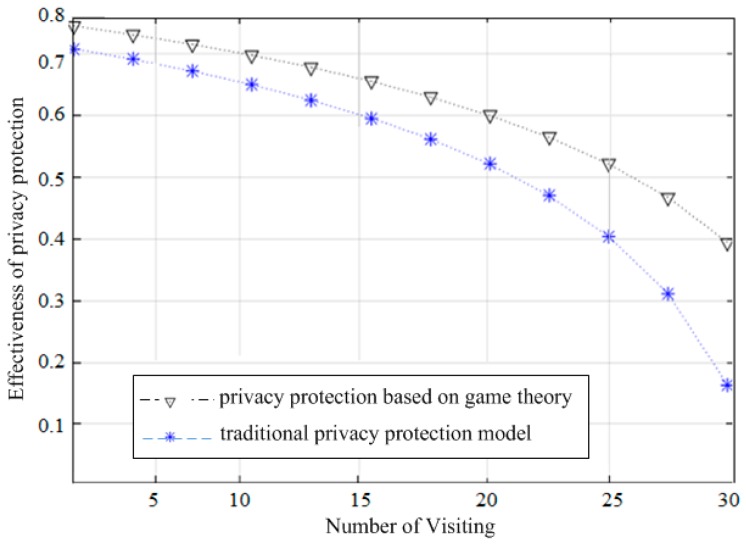
Comparison of the privacy protection between the traditional and game-based protection models.

**Figure 6 sensors-19-01581-f006:**
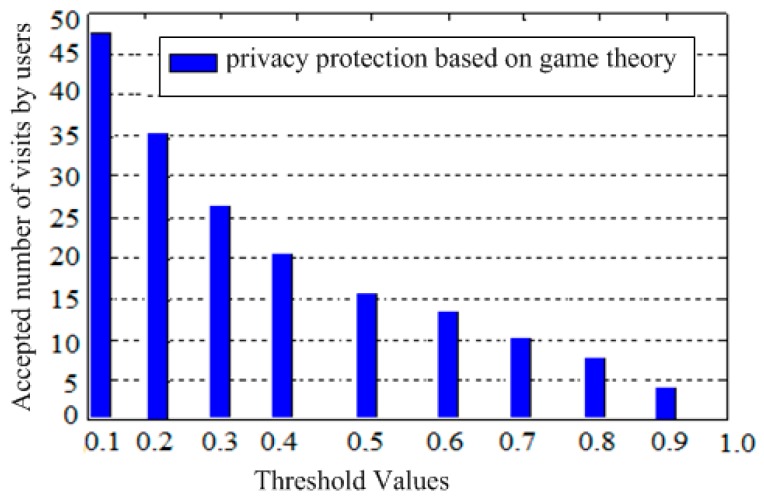
Relationship between the threshold values and tolerance of game theory-based privacy protection model.

**Table 1 sensors-19-01581-t001:** Analyzing the reward matrix in the traditional privacy protection model.

Owners (Users)	Third-Party Visitors	
	Good-Faith	Malicious
Allow	$Q_{in}{}_{wel}^{acp},\;B_{in}{}_{wel}^{acp}$	$-Q_{loss}{}_{mal}^{acp}$,$\;B_{in}{}_{mal}^{acp}$
Refuse	$-Q_{in_{wel}^{acp}}$, 0	$Q_{in}{}_{mal}^{den}$, 0

**Table 2 sensors-19-01581-t002:** Relationship between the degree of private information leakage and threshold values.

Tolerance	Threshold
Very High	[0, 0.2]
High	(0.2, 0.4]
Medium	(0.4, 0.6]
Low	(0.6, 0.8]
Very Low	(0.8, 1]

**Table 3 sensors-19-01581-t003:** Analyzing the reward matrix in the game-theoretic privacy protection model.

Owners (Users)	Third-Party Visitors	
	Good-Faith	Malicious
Allow	$Q^{\prime }{}_{in}{}_{wel}^{acp},\;B^{\prime }{}_{in}{}_{wel}^{acp}$	$-Q^{\prime }{}_{loss}{}_{mal}^{acp}$, $B^{\prime }{}_{in}{}_{mal}^{acp}$
Refuse	$-Q^{\prime }{}_{in_{wel}^{acp}}$, 0	$Q^{\prime }{}_{in}{}_{mal}^{den}$, $-B^{\prime }{{}_{loss}}_{mal}^{den}$

**Table 4 sensors-19-01581-t004:** Number of visits and private information leakage probability.

No. Visits.	Private Information Leakage Probability	
	Traditional Model	Game Theoretic Model
1	0.05	0.01
2	0.05	0.03
3	0.05	0.04
4	0.08	0.06
5	0.14	0.08

**Table 5 sensors-19-01581-t005:** The relationship between the threshold imposed by the user and the number of access results.

Threshold	Number of Visits
0.1	52
0.2	40
0.3	33
0.4	28
0.5	0.5
0.6	15
0.7	10
0.8	6
0.9	3
1.0	0

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
