# Peer review of "A Game-Theoretic Framework to Preserve Location Information Privacy in Location-Based Service Applications"

_sensors, 2019, doi:10.3390/s19071581_

Round 1
Reviewer 1 Report
Summary:
The paper "A Game-Theoretic Framework to Preserve Location Information Privacy in Location-based Service Applications" deals with
privacy in the context of location-based services. The authors describe the latter setting extensively in the light of smart mobile
devices and location-based services. In particular, they discuss the existing technical-oriented mechanisms to preserve privacy of a user
that utilizes location-based services and potential intruders as well as their interests. In contrast to existing solutions, the authors
propose a framework based and inspired from the field of game theory. Their framework shall help users to deal with privacy concerns by
giving location-based service users and their "opponents" a way to negotiate on the access of the location of an user based on gaming aspects
and incentives. In this context, traditional game theory aspects are investigated, e.g., the Nash equilibrium. A formal framework is presented
as well as experimental results, in which the approach of the authors is compared to other existing solutions.
Points in favour:
- The paper fits to the scope of the journal
- The paper has a clear structure
- The contribution of the paper is well elaborated
- The paper shows experimental results
- The technical part is sound
- The paper discusses related work properly
- The paper deals with a topic that is contemporary and also addressed by the relevant communities
Points against the paper:
- The paper needs a revision on the language and writing
e.g.,
(1) Abstract first sentence is grammatically wrong, verb is missing
(2) Line 13: self-motive -> does not exist
(3) Line 22: in to account -> into account
(4) Line 35: when location information has to .... -> makes no sense
(5) in general: LBSs -> not common
(6) Line 48: ... location information on obfuscated or anonymized ... -> makes no sense
...
The paper needs a language revision, the content is well crafted
Author Response
Dear Reviewer
Thank you so much for reviewing our manuscript. We also greatly appreciate for your complimentary comments and suggestions. Please find our point-by-point responses in (RED) in the file attached.
Thanks again!
With best regards,
Dr.Mulugeta Kassaw on behalf of co-authors

Reviewer 2 Report
The authors propose a game-theoretical technique which strikes a balance between the amount of information a user is comfortable to share and the amount of information an adversary can learn in location-based services.
The overall idea is interesting and new, to the best of the reviewers' knowledge. The presentation of the material, as well as the related work, is adequate but the writing can be much improved. There are many typos. Please, have the manuscript proofreading.
While the reviewer understands the importance of improving privacy in location-based services, there is a concern in the application of the proposed technique. There seems to be a transient time in which the users' data is exposed to the adversary (until the equilibrium is reached?) The user needs to be in a location multiple times before the proposed technique is effective. There is a dependence from a threshold, which concerns the reviewer the most as it is not entirely clear how the value of such a threshold is related to the information that the user is willing to share. More importantly, it is not entirely clear how a user would set it up. This kind of considerations should be explained in the manuscript. What would be the criteria that the informed rules for the user to set the threshold? Is there a way to assign the threshold value automatically?
Experimental results are interesting as they are based on real-world data, but the actual experiments are simulated and as such artificial. It is not clear to the reviewer how the proposed technique would work in practice. The authors have a system level description of the proposed framework. Is this framework implemented in a mobile device? Do the authors plan to run real experiments? It would be good to see how actual experiments match simulation results.
Author Response

(The authors gave the same response as above.)
